# Solvent induced amyloid polymorphism and the uncovering of the elusive class 3 amyloid topology
Zsolt Dürvanger [1,2], Fruzsina Bencs [1,3], Dóra K. Menyhárd [1,2], Dániel Horváth [1,2] & András Perczel [1,2] ✉

Aggregation-prone-motifs (APRs) of proteins are short segments, which – as isolated peptides - form diverse amyloid-like crystals. We introduce two APRs - designed variants of the incretin mimetic Exendin-4 - that both display crystal-phase polymorphism. Crystallographic and spectroscopic analysis revealed that a single amino-acid substitution can greatly reduce topological variability: while LYIQWL can form both parallel and anti-parallel β-sheets, LYIQNL selects only the former. We also found that the parallel/anti-parallel switch of LYIQWL can be induced by simply changing the crystallization temperature. One crystal form of LYIQNL was found to belong to the class 3 topology, an arrangement previously not encountered among proteinogenic systems. We also show that subtle environmental changes lead to crystalline assemblies with different topologies, but similar interfaces. Spectroscopic measurements showed that polymorphism is already apparent in the solution state. Our results suggest that the temperature-, sequence- and environmental sensitivity of physiological amyloids is reflected in assemblies of the APR segments, which, complete with the new class 3 crystal form, effectively sample all the originally proposed basic topologies of amyloid-like aggregates.

Amyloids are fibrillar nanostructures composed of highly ordered assemblies of polypeptides with a characteristic cross-β architecture[1]. As amyloid aggregation is implicated in more than fifty human diseases including Alzheimer's and Parkinson's disease[2,3], amyloid fibrils have traditionally been considered to be a "toxic" molecular structure. However, beside the pathological self-assembly of misfolded globular or intrinsically disordered proteins, examples for functional amyloid formation can also be found: certain physiologically active (poly)peptides and proteins utilize amyloid aggregation to perform their roles[4–6]. The aggregation process itself that creates fibrils with such extraordinary rigidity, such complex molecular architecture and chemical stability, has inspired various materials science applications and the creation of artificial amyloids too[7–9], serving a wide array of industrial purposes, ranging from water purification to 3D printing[10]. Within the sequences of amyloidogenic proteins, short segments of 5-15 residues, called aggregation prone regions (APRs), can be identified which possess a high propensity for amyloid aggregation[11], and were proposed to initiate[12] and/or mediate amyloid formation of full-length proteins[13,14]. APRs carry recognizable traits, so much so that numerous different algorithms were developed for the identification and scoring the aggregation-power of APR motifs in protein sequences[15–17,12].

As APRs are able to initialize the aggregation of longer sequences, in naturally occurring folded proteins carrying APR segments various means of protection against unwanted amyloid formation can be identified. Forcing an APR into an α-helical structure is known to be a widespread "strategy" of nature to prevent aggregation[18]. In fact, it has been suggested that those segments that are predicted to favour the β-sheet secondary structure, but are "transiently locked" into an α-helix, will probably be amyloidogenic[19]. Alternatively, amyloidogenic sequences can be seen buried within the hydrophobic core of the protein or shielded by loops or by multimerization[20], thereby hindering the access to these reactive, or "sticky" β-edges. The presence of charged (Asp, Glu, Lys, Arg) or conformationally restricted (Pro) amino acids that are incompatible with a typical β-sheet structure - often referred to as gatekeeper residues - was found to be yet another means of protection against aggregation[6,21–23].

The first atomic resolution insights into the cross-beta spine structure of amyloids were obtained from the amyloid-like crystals of 6-8 residue long APR segments of amyloidogenic proteins[24,25]. Since then, more than 170 high-resolution crystal structures of APRs have been determined using X-ray and electron diffraction methods. In addition to being a model for amyloidogenic core regions of proteins, the amyloid formation of smaller

---

[1]Laboratory of Structural Chemistry and Biology, ELTE Eötvös Loránd University, Pázmány Péter sétány 1/A, H-1117 Budapest, Hungary. [2]HUN-REN-ELTE Protein Modeling Research Group, ELTE Eötvös Loránd University, Pázmány Péter sétány 1/A, H-1117 Budapest, Hungary. [3]Hevesy György PhD School of Chemistry, ELTE Eötvös Loránd University, Pázmány Péter sétány 1/A, H-1117 Budapest, Hungary. ✉e-mail: perczel.andras@ttk.elte.hu

oligopeptides or even single amino acids[26,27] provide small, tractable model systems for studying the aggregation process itself and the key interactions that stabilize the unique amyloid phase. Recent studies even suggest that flat-ribbon like amyloid arrangement that the crystal structures of APRs also belong to, may represent the absolute minimum of the folding free energy landscape of proteins, providing further significance to better our understanding of the basic properties and the mechanisms that lead to the formation of such crystals[10,28,29].

The common feature of all amyloid crystal structures is the arrangement of the polypeptide chains into a cross-β structure. Eisenberg et al. originally classified amyloid-like crystal structures into 8 topological classes, based on the relative direction of neighbouring β-strands[24]. This classification system was later extended to 10 classes in order to include all of the theoretically possible arrangements[1,30] (Fig. 1). While no examples have yet been observed for the two additional classes (classes 9 and 10), several examples have been reported for most of the eight classes originally proposed, with the exception of class 3. The sole crystal structure containing exclusively class 3 steric zipper interfaces is formed by a peptide built of non-proteinogenic amino acids[31], making it less relevant from a biological perspective. Class 3 homo-zipper interfaces have been identified in the crystal structure of the SVQIVY peptide (PDB code: 6ODG[32]), and were confirmed to be present in the amyloid fibrils of the ISFLIF segment by solid state NMR[33], but in these cases alternatingly appearing interfaces that belong to different topologies (class 1 in case of the former and class 2 in case of the latter) also contribute to the stabilization of the amyloid-like structure. To the best of our knowledge, a crystal structure containing exclusively class 3 homo-zipper interfaces formed by natural amino acids has not been previously observed.

The class 3 amyloid topology has the peculiarity of having the same amino acids facing each other in the dry-zipper interface (Fig. 1). This is an important distinction, separating class 3 from class 2 arrangements, where the relative direction of the neighbouring β-strands is the same, but the two facing sides of the dry zipper interface have different chemical compositions. This requirement of complete self-complementarity that also involves the unfavourable electrostatic interactions of the same-charged chain-termini is

the most plausible explanation for the relative rarity of the class 3 topology among amyloid crystal structures.

While there is increasing evidence suggesting that any primary sequence can lead to amyloid-like aggregation under the right conditions[34], the propensity for amyloid formation and the exact structural features of amyloid aggregates – similarly to the folding topologies of globular proteins – are highly sequence dependent[35]. Much effort has been devoted to understanding the reasons behind the selection of certain amyloid topologies, as the relative orientation of adjacent β-sheets is an essential feature of peptide-assembled amyloid systems[36–38]. Understanding the relationship between the primary sequence of amyloidogenic regions and the specific structural motifs they form is necessary to avoid the uncontrolled amyloid formation of polypeptides and proteins, for which the easily adaptable and versatile APRs provide ideal model systems. Both the pharmaceutical industry and the design of novel materials that exploit amyloid-like self-assembly processes rely on this key structural information.

Several algorithms have been developed to predict favourable arrangements of peptides in amyloid assemblies[36,39]. However, the success of structure-based prediction of amyloidogenicity and amyloid topologies relies significantly on the number and conformational/topological diversity of high-resolution amyloid structures determined. From this point of view, amyloid structures with novel conformational or topological features are of particular importance, as they may expand the learning space available for structure-based prediction algorithms.

We previously characterized the APR of the Tc5b miniprotein – LYIQWL –, and found that it is able to form polymorphic amyloid crystal structures with both parallel and antiparallel β-sheets[6,40]. In these structures the side chain of Trp5 adopted a variety of different conformations, suggesting that this residue might play a role in the amyloid polymorphism of LYIQWL. We asked the question, whether a single point mutation of Trp5 would be enough to make the parallel orientation of the β-sheets more favourable. We selected asparagine, as its side chain is less bulky than tryptophan and it was shown to stabilize parallel β-sheets by forming hydrogen bonded ladder-motifs[41]. In the present work we therefore synthesized the Trp5Asn mutant of LYIQWL and found that it forms a purely

**Fig. 1 | The ten symmetry classes of amyloid-like crystal structures.** Peptide monomers adopt an extended β-strand secondary structure (indicated by arrows pointing from the N-terminus to the C-terminus) in all amyloid-like crystal structures that assemble into parallel (classes 1–4 and 9–10) or antiparallel (classes 5–8) β-sheets. In the β-sheet secondary structure the side chains of even-numbered residues (face, shown in blue) face in the opposite direction than that of the odd-numbered residues (back, shown in red). The β-sheet layers are connected to each other by favourable side-chain interactions through steric zipper interfaces. Depending on the relative orientation of the β-strands, interfaces can form between the same (face-to-face) or the opposite (face-to-back) surfaces of the β-strands. In the parallel classes, two β-sheets meeting at the steric zipper interface can either run in the same direction (N-to-N), or in the opposite direction (N-to-C). In classes 7 and 8, the same faces of the β-strands in adjacent layers run in the same direction (class 7, up-up) or in the opposite direction (class 8, up-down).

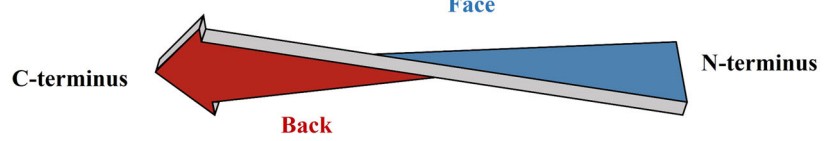

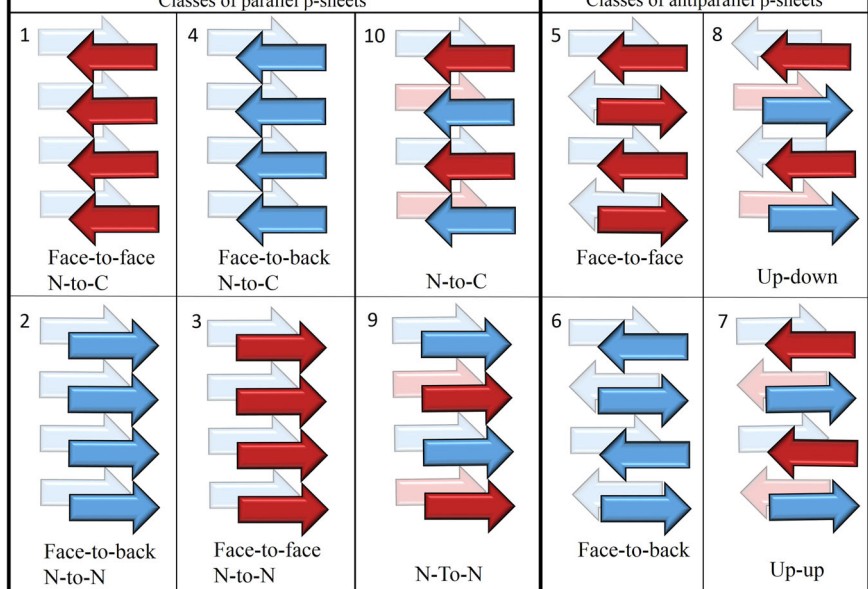

**Fig. 2 | Polymorphic amyloid crystal structures of LYIQWL.** Crystals grown from water (**a**), 10 v/v% ethanol in the presence of TFA (trifluoroacetic acid) (**b**), 30 v/v% ethanol (**c**) and 10 v/v% ethanol in the absence of TFA (**d**). Water molecules are represented by red spheres, ethanol and TFA molecules are shown by green and grey sticks, respectively. Crystals grown from water and 10 v/v% ethanol in the presence of TFA consist of antiparallel β-sheets (class 8 amyloid topology), whereas crystals grown from 30 v/v% ethanol and 10 v/v% ethanol in the absence of residual TFA consist of parallel β-sheets of class 1 and class 4 topology, respectively. Distinct steric zipper interfaces in the two parallel structures are labelled as S1, S2, S3 and S4. **e** Comparison of the conformation of individual chains in the polymorphic structures of LYIQWL. Structures from the present work are marked with*.

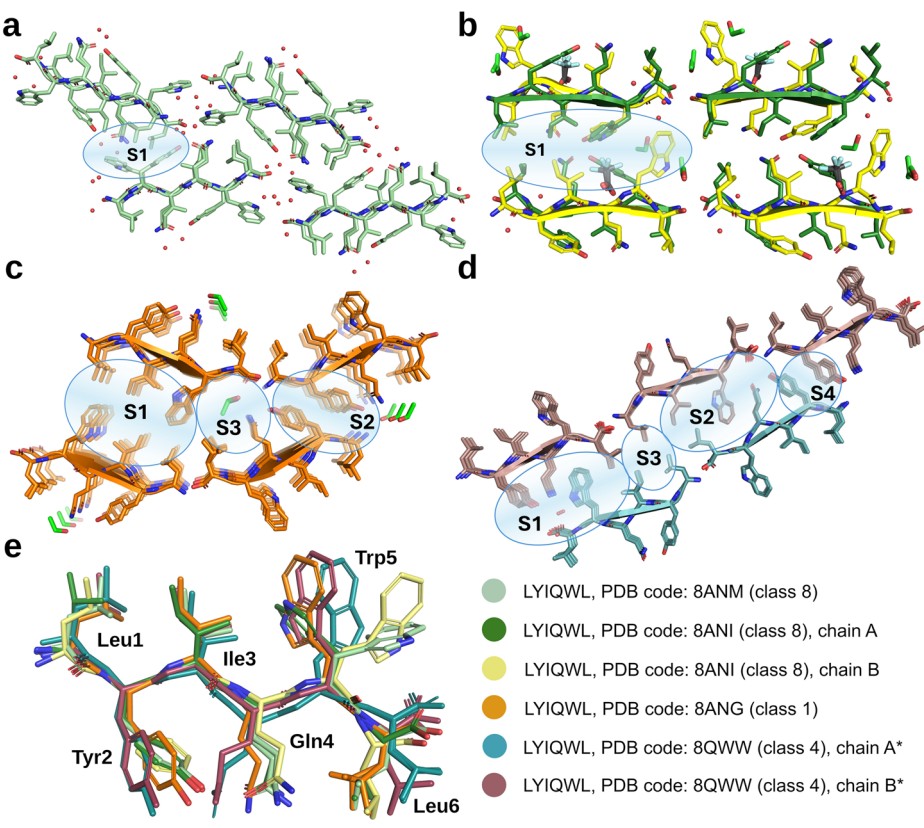

○ LYIQWL, PDB code: 8ANM (class 8)
● LYIQWL, PDB code: 8ANI (class 8), chain A
○ LYIQWL, PDB code: 8ANI (class 8), chain B
● LYIQWL, PDB code: 8ANG (class 1)
● LYIQWL, PDB code: 8QWW (class 4), chain A*
● LYIQWL, PDB code: 8QWW (class 4), chain B*

class 3 amyloid crystal structure. To further characterize its amyloid assemblies both in the solid phase and in solution, we studied the peptide with Atomic Force Microscopy (AFM), Electronic Circular Dichroism spectroscopy (ECD) and Fourier-transform Infrared Spectroscopy (FTIR). By comparing the results with its parent hexapeptide, LYIQWL, we found that the Trp5Asn mutation created a preference for parallel amyloid structures. Furthermore, comparison of the crystal structures with the spectroscopic results highlighted similarities between amyloid structures in the crystalline and in the solution phases.

## Results
### Solvent and temperature effects in crystals of LYIQWL
We previously found that unlike the majority of amyloid forming peptides, LYIQWL was able to crystallize both in parallel and antiparallel β-sheets, depending on the crystallization conditions (Fig. 2a–c). Specifically, crystals containing antiparallel β-sheets (class 8 topology, Fig. 2a, b) were obtained from water and 5–10 v/v% ethanol, while crystals grown from 20–30 v/v% ethanol contained parallel β-sheets (class 1 topology, Fig. 2c). Although all crystal structures of LYIQWL contained the peptide chains in an extended β-strand conformation (Fig. 2e), significant differences could be observed in the conformation of side chains, allowing for different interaction patterns in the different structures[6].

In crystals grown from 5–10 v/v% ethanol, TFA (trifluoroacetate) ions – present as a minor contaminant from the peptide purification process – could be identified in the structure (shown as grey sticks in Fig. 2b (PDB code: 8ANI)[6]), raising the question of whether residual TFA might have influenced the amyloid self-assembly or crystal formation of the peptide in all other cases as well. To study the potential interference of TFA, we have lyophilized the peptide in HCl, and crystallized it again from 10 v/v% ethanol. Within a few weeks, thick, needle-like crystals grew in clusters at both 4 °C and 37 °C. Crystals grown at 4 °C contained antiparallel β-sheets and were found to be identical to those obtained from water at 4 °C or even at 37 °C (in the presence of possible TFA contamination) (Fig. 2a (PDB code: 8ANM)[6]), indicating that the presence or absence of small amounts of

TFA or ethanol does not necessarily induce a change in the mode of self-assembly. Interestingly, crystals grown at 37 °C (in the absence of TFA) consisted of parallel β-sheets (Fig. 2d). In this case a simple temperature change lead to an anti-parallel/parallel switch: starting solutions of completely identical chemical composition resulted in amyloid-like crystalline phases of entirely different topology. Similar structure determining effect of temperature was seen recently in case of prion amyloid fibrils of more than 140 residue length[42], demonstrating yet again, how surprisingly well the crystal phase ordering of these small model peptides reflect the aggregation processes of much longer, physiological variants.

The crystals grown at 37 °C from 10 v/v% ethanol also differed from those previously grown from 20–30 v/v% ethanol in the presence of residual TFA at 20 °C or 37 °C (Fig. 2c (PDB code: 8ANG)[6]) in both the conformation of the individual chains and the type of the amyloid topology. In crystals obtained from a solution containing more ethanol (and possibly TFA), the neighbouring β-sheets meet at an angle at the chain termini (Fig. 2c) giving rise to three different interfaces (S1, S2 and S3 in Fig. 2c), of which the largest (S1) corresponds to the class 1 topology, while S2 and S3 are class 4 interfaces. Because the class 1 interface is more extended and dominates the overall arrangement, we classified this crystal form as belonging to the class 1 topology. On the other hand, in crystals from the present work, grown from a solution containing less ethanol and no residual TFA, neighbouring β-sheets run parallel to each other, resulting in two larger class 4 (S1 and S2 in Fig. 2d) and two smaller class 1 interfaces (S3 and S4 in Fig. 2d) leading to an overall class 4 topology.

The hydrophobic residues (Leu1, Ile3, Trp5) form clusters in both structures, while hydrogen bonds are formed by chain termini, solvent molecules, Tyr2, Gln4 and Trp5 of chain A in the class 4 structure. A common structural feature in the two parallel LYIQWL structures is the presence of a weak Tyr – Tyr interaction (S2 in Fig. 2c and S4 in Fig. 2d). The geometric parameters characterising this interaction are similar in the two structures: the distance between the centres of the aromatic rings are 5.6 Å and 6.1 Å in the class 1 and class 4 structures, respectively, while the angle between the planes of the rings are 81.9° and 86.7°.

**Fig. 3 | Polymorphic amyloid crystal structures formed by LYIQNL.** The structure of monoclinic crystals grown in the presence of ethanol and displaying the class 3 topology are shown in blue and light red, while the structure of orthorhombic crystals grown without ethanol (class 4) is shown in olive. Both the crystals grown in the presence (**a**) and in the absence of ethanol (**b**) contained β-strands of parallel orientation. **c, d** Interfaces and solvent molecules in the two crystal structures. Distinct zipper interfaces are labelled with S1 and S2. Water and ethanol molecules are shown as red spheres and green sticks, respectively. **e, f** Schematic representation of the packing topologies in the two crystals. **g** Comparison of the peptide chains found in the two crystal structures of LYIQNL.

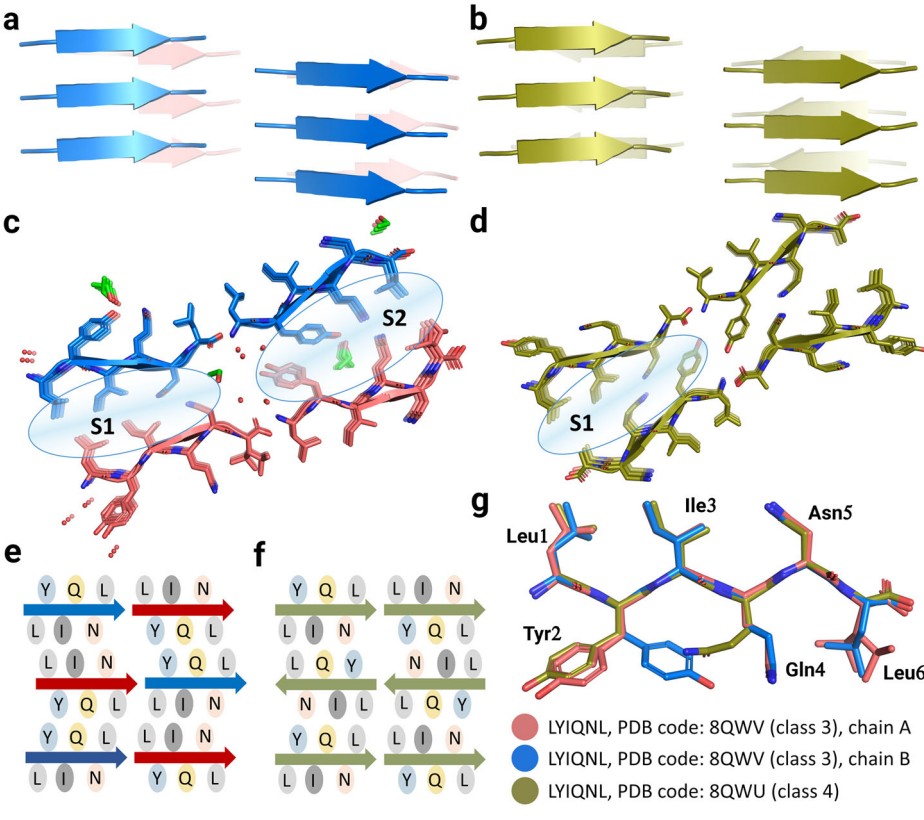

In contrast to the class 1 structure, the class 4 structure did not contain ethanol molecules – only a water molecule could be identified in the electron density map. This suggests that the presence of an increased amount of ethanol is able to induce an anti-parallel to parallel reorientation of β-sheets, even without forming direct ethanol-peptide contacts within the amyloid nanostructure.

## Polymorphic crystal structures of LYIQNL reveal the class 3 amyloid topology

In order to determine whether LYIQNL is capable of forming amyloid structures in its crystalline form, we undertook crystallization from a variety of conditions. Needle-like crystals grew from 0.1 M acetate buffer, both in the presence and absence of ethanol at pH=4.8 and diffracted to a resolution of 1.70 Å and 1.55 Å, respectively. Crystals grown from both solvents contained β-strands of parallel orientation (Fig. 3a, b), but arranged in different amyloid topologies (Fig. 3c, d). In the monoclinic crystals grown from the ethanol containing solvent system, strands arranged in the class 3 topology could be observed (Figs. 1 and 3c, e), while the orthorhombic crystals grown in the absence of ethanol contained strands arranged according to the class 4 topology (Figs. 1 and 3d, f).

While the class 4 structure did not contain any solvent molecules, the class 3 structure contained both water molecules and two ethanol molecules, one in a solvent channel near the chain termini and another wedged into the steric zipper interface (Fig. 3c, d).

LYIQNL is one of the few amyloidogenic oligopeptides that has been reported to form multiple polymorphic crystal structures. Since the atomic level arrangement determines the macroscopic features of amyloid fibrils, it is also crucial to understand the factors that prevent or promote polymorphism. Despite the completely different packing of the chains, similar interfaces form within the two polymorphic crystal structures of LYIQNL. In the class 3 topology (monoclinic crystal) the same faces of the chains interact with each other, thereby forming two distinct zipper interfaces (Fig. 3c, e), namely that of Leu1, Ile3, Asn5 of chain #1 with Leu1, Ile3, Asn5 of chain #2 (S1 in Fig. 3c), as well as Tyr2, Gln4, Leu6 of chain #1 with Tyr2,

Gln4, Leu6 of chain #2 (S2 in Fig. 3c). The assembly is stabilized by a hydrophobic cluster formed with Leu and Ile side chains (Fig. 3c, e) complemented by "pure" Asn-Asn and Gln-Gln H-bond pairs (Fig. 4c). Finally, ethanol molecules insert themselves into the zipper interface through the formation of multiple H-bonds with both Gln and Tyr side chains (Fig. 3c).

In the class 4 topology on the other hand, opposite faces of the oligopeptide form a single type of steric zipper interface, that is, the two interfaces formed by the neighbouring peptide chains are identical (Leu1, Ile3 and Asn5 of chain #1 interact with Tyr2, Gln4 and Leu6 of chain #2, interface S1 in Fig. 3d). Similarly to the class 3 structure, hydrophobic residues (Leu and Ile) form a hydrophobic niche in this arrangement too, while polar side chains of Asn and Gln stabilize the interface enforced by H-bonds (Fig. 3f and Fig. 4d). The side chain of Tyr2 forms H-bonds with chain termini (Fig. 3d and Fig. 4b). The two polymorphic structures with distinctly different topologies are thereby stabilized by very similar interaction patterns.

Interactions between aromatic residues were shown to play a key role in the amyloid formation of several oligo- and polypeptides[43–46]. We therefore analysed the interactions of aromatic side chains (Tyr2) in the two crystal forms. Due to the parallel orientation of the β-strands, Tyr ladders with a distance of 4.8 Å between adjacent side chains are formed in both crystal structures. This results in a favourable, displaced face-to-face interaction mode[47,48]. Tyr side chains also interact within the steric zipper interface in the class 3 crystal structure, here again with a displaced face-to-face geometry (Fig. 4a). On the other hand, in the class 4 arrangement, the Tyr side chains face in the same direction, so that inter-β-sheet Tyr-Tyr interactions could not form at the zipper interface. Although the Tyr side chains also meet near the chain ends in this crystal form, the interplanar distance between them is 7.0 Å, indicating a much weaker, if any, type of interaction (Fig. 4b).

While the backbone conformation of LYIQNL is almost identical in the two all-parallel β-sheet amyloid polymorphs we found, slight differences in the side-chain orientations of Tyr2 (two rotamers in the class3, and a single rotamer in the class4 crystal form) and Gln4 (differing orientations) can be seen, creating more favourable local environments (Fig. 3g).

**Fig. 4 | Aromatic π-π interactions and H-bond networks stabilizing the dry-zipper interfaces of LYIQNL.** The monoclinic crystal form is comprising a pure class 3 amyloid crystal structure (**a**, **c**), while the orthorhombic crystal form incorporates a class 4 amyloid topology (**b**, **d**).

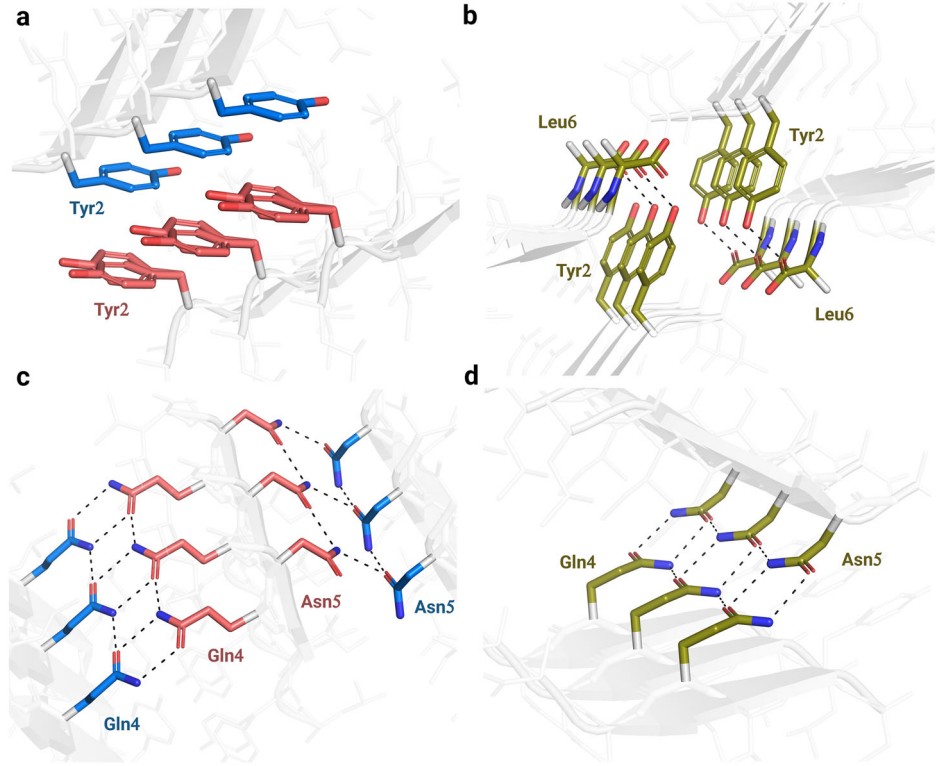

## Polymorphism is enabled by the similar energetics of different β-sheet arrangements

In order to gain a better understanding of the polymorphic nature of the two sequences, we studied the energetics of their amyloid formation using the determined crystal structures. We calculated both the energy gain upon formation of a "double layer" along a given interface from individual β-sheets using atomic solvation parameters[49] and the free energy of formation of the same "double layer" from monomeric peptide chains using the method proposed by Eisenberg et al.[50] (Table 1).

The energy gain upon formation of a hypothetical double layer from individual β-sheets is comparable to that of found for stable steric zippers (median = −1.417 kcal/mol/strand[49]) and significantly higher than that of found for LARKS structures (median = −0.443 kcal/mol/strand[49]) by Hughes et al. Lower values were found in case of 8ANM where a small interface with small buried area formed and in S2 from 8QMV, where an ethanol molecule is part of the interface.

**Table 1 | Energetics of formation of a hypothetical double layered amyloid structure along given interfaces found in the crystal structures**

| Sequence | Polymorph | Interface | $E_{form}$ (from sheets) [kcal ·mol⁻¹/ strand] | $E_{form}$ (from monomers) [kcal ·mol⁻¹/strand] |
|---|---|---|---|---|
| LYIQWL | 8ANM (class 8) | S1 | −0.85 | −5.19 |
| | 8ANI (class 8) | S1 | −1.54 | −4.58 |
| | 8ANG (class 1) | S1 | −1.66 | −5.31 |
| | 8QWW (class 4) | S1 | −1.30 | −4.56 |
| | | S2 | −1.37 | −4.63 |
| LYIQNL | 8QWV (class 3) | S1 | −1.17 | −3.56 |
| | | S2 | −0.76 | −3.21 |
| | 8QWU (class 4) | S1 | −1.23 | −3.80 |

Furthermore, both measures – but especially the energy of formation from monomers – show slightly more favourable energetics in case of LYIQWL than of LYIQNL, while the energy difference between different polymorphs of the same peptide sequence is decidedly lower. Since the atomic solvation parameters were optimized using water as a solvent while several of the polymorphs discussed here were observed only in the presence of ethanol, some error might have been introduced into our estimates, but the results clearly reflect that polymorphism is simply enabled by the possibility of forming assemblies of similar energies with different structural characteristics.

## LYIQWL and LYIQNL form β-sheet rich assemblies in solution

To investigate the progress of amyloid self-assembly we used ECD and FTIR spectroscopic methods. Both techniques faithfully monitor the earliest phase of β-strand formation and self-stabilisation in solution. Shifts in the vibrational and chiroptical bands characterise the initial nanostructures (1–100 nm). The solvent dependence of the self-assembly was studied in two different solvents: pure water and 10 v/v% ethanol at 37 °C. The elevated temperature and the presence of ethanol reduces the permittivity (dielectric constant) of the solvent (Supplementary Table 1). To characterize the macroscopic amyloid assemblies of the two peptides, we used AFM, which can provide information on the further maturation of amyloid crystals and/or filaments in a larger size range (1–100 μm).

LYIQWL adopts a β-sheet rich secondary structure in water as evidenced by characteristic ECD and FTIR spectral changes and AFM images (Table 2 and Fig. 5a–c). Consistent with amyloid fibril formation, the amide I band of LYIQWL shifts to 1629 cm⁻¹ within 48 h (Fig. 5a)[51,52] complemented by the appearance of a band at 1690 cm⁻¹, indicating that these β-sheets are antiparallel[51,53] (Supplementary Fig. 1). Furthermore, both ECD and IR spectral changes indicate the presence of aromatic interactions. The increase in intensity of the band at 1518 cm⁻¹ in the IR spectrum indicates that the nascent amyloid structure is solidifying. This may be due to the π-π interaction of the Tyr2 side chains. (The vibrational band resulting from the restricted motion of Trp is expected to be weak in intensity, overlapping with the vibrational frequencies of Tyr ν(CC) and δ(CH)[54].) A strong but transient

**Table 2 | Summary and interpretation of the observed spectral signals of LYIQWL and LYIQNL in water and 10 v/v% ethanol containing solvents**

| Sequence | Solvent | IR observation | ECD observation | Conclusion |
|---|---|---|---|---|
| LYIQWL | 0 v/v% ethanol (100 v/v% H$_2$O) | amide I band shifts $\nu$(C=O): 1636 cm$^{-1}$ (0 h) →1629 cm$^{-1}$ (48 h) | spectrum characteristic to β-sheet (48 h) | formation of amyloid nanostructures[51,52] |
| | | amide I sideband $\nu$(C=O) appears at 1690 cm$^{-1}$ (6 h) | | presence of antiparallel β-sheets[51,53] |
| | | increased intensity of Tyr–OH $\nu$(CC), $\delta$(CH): 1518 cm$^{-1}$ | | fixed position of Tyr2[54] |
| | | | strong, transient couplet at 225 nm and 237 nm $\pi\rightarrow\pi^*$ transitions ($^1$L$_b$) | transient aromatic interaction during amyloid formation[40,75] |
| LYIQNL | | amide I band shifts $\nu$(C=O): 1640 cm$^{-1}$ (0 h) →1632 cm$^{-1}$ (48 h) | spectra characteristic to β-sheets (48 h) | formation of amyloid nanostructures |
| | | absence of amide I sideband $\nu$(C=O) at 1690 cm$^{-1}$ (48 h) | | presence of parallel β-sheets |
| | | increased intensity of Gln and Asn $\nu$(C=O) band at 1658 cm$^{-1}$ (48 h) | | fixed position of Asn/Gln side chains[54], formation of Asn/Gln ladders |
| LYIQWL | 10 v/v% ethanol (+ 90 v/v% H$_2$O) | amide I band $\nu$(C=O): 1629 cm$^{-1}$ (48 h) | spectra characteristic to β-sheets | rapid formation of amyloid nanostructures |
| | | absence of amide I sideband $\nu$(C=O) at 1690 cm$^{-1}$ (48 h) | | presence of parallel β-sheets |
| | | | absence of transient couplet observed in water (0–48 h) | different aggregation mechanism in the two solvents |
| | | | positive band at 222 nm | weak aromatic interaction, different from the transient interaction detected in water |
| LYIQNL | | amide I band $\nu$(C=O): 1632 cm$^{-1}$ (48 h) | spectrum characteristic to β-sheet (48 h) | formation of amyloid nanostructures |
| | | increased intensity of Gln and Asn $\nu$(C=O) band at 1657 cm$^{-1}$ (48 h) | | fixed position of Asn/Gln side chains, formation of Asn/Gln ladders |
| | | increased intensity of Tyr–OH $\nu$(CC), $\delta$(CH) at 1517 cm$^{-1}$ | intensive positive band at 190 nm $\pi\rightarrow\pi^*$ transitions ($^1$B$_a$ + $^1$B$_b$) | fixed position of Tyr2, presence of aromatic π-π interactions[75] |

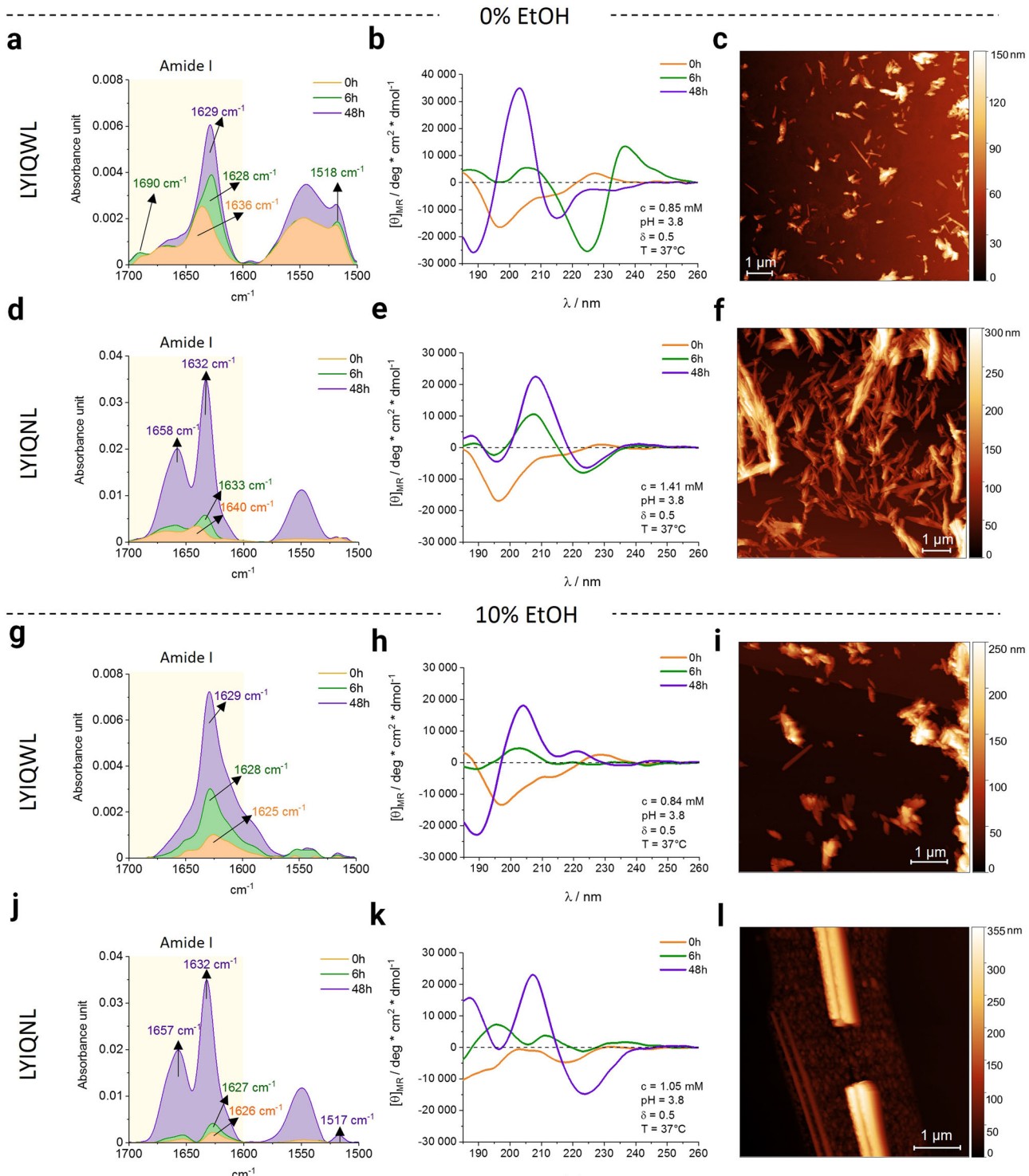

**Fig. 5 | The time-dependent FTIR (first column) and the time-dependent ECD spectra (second column) with the AFM images (third column) taken after 48 h mixing of the LYIQWL and LYIQNL peptides.** The amide I bands at 1629 cm⁻¹ and 1632 cm⁻¹ (**a, d** green) indicate β-sheet formation in water of LYIQWL and LYIQNL, respectively. The band at 1690 cm⁻¹ (**a**) shows that LYIQWL adopts an antiparallel β-strand arrangement, while the absence of this band shows that LYIQNL adopts a parallel β-strand arrangement within the nascent, but still soluble amyloid. Both peptides are unstructured at t = 0 h, (**b, e**) U-type spectra: orange; and subsequently form a β-strand (**b, e**). B-type spectra: green. The transient strong exciton couplet (pair of signals at 225 nm and 237 nm, **b** green) arises from a π-π interaction, whose transient signal interferes with the rising B-type ECD spectra of the nascent amyloids. (This transient effect was captured for LYIQWL, but not for LYIQNL.) After incubation and mixing at 37 °C, rudimentary fibril-like structures were formed, which are characteristic of the amyloid structure (**c, f**). From (**g–l**) spectra and images were recorded in 10 v/v% ethanol. In the presence of ethanol neither the transient π-π interaction, nor the vibrational transition characteristic of an antiparallel β-sheet (1690 cm⁻¹) was detected (**g, j**). However, for both peptides a B-type ECD spectrum is obtained after 6 h, which becomes convincing after 48 h, confirming the rise of the amyloid structure (**h, k**). This increase in solubilized amyloid in 10% ethanol is also confirmed by the FTIR spectrum and AFM images (**i, l**).

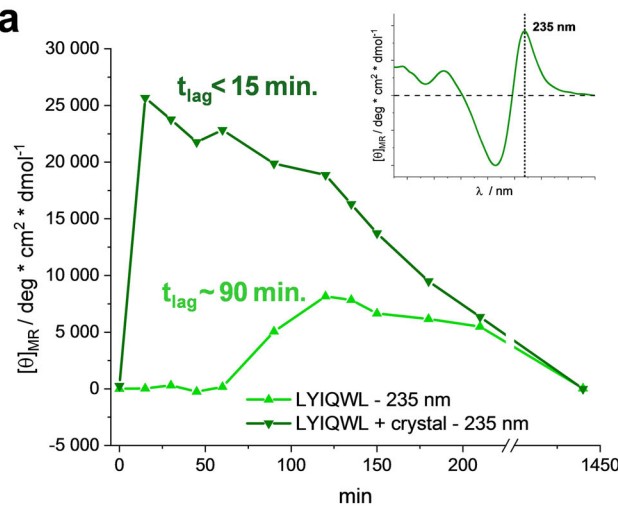

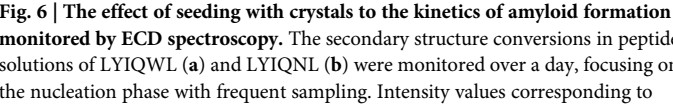

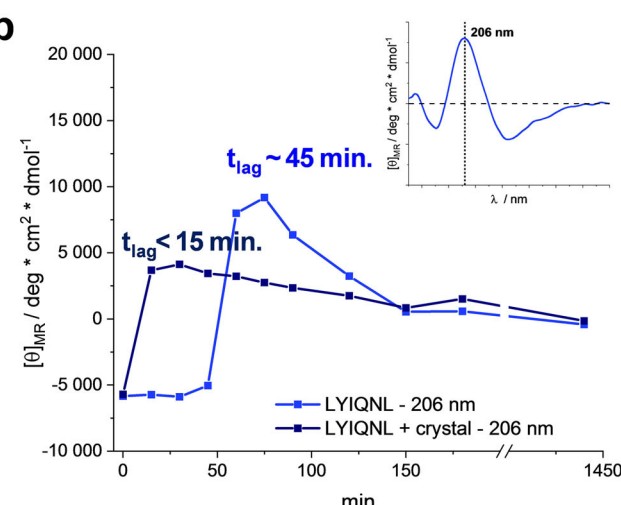

**Fig. 6 | The effect of seeding with crystals to the kinetics of amyloid formation monitored by ECD spectroscopy.** The secondary structure conversions in peptide solutions of LYIQWL (**a**) and LYIQNL (**b**) were monitored over a day, focusing on the nucleation phase with frequent sampling. Intensity values corresponding to wavelengths characteristic of the β-sheeted state for LYIQWL (235 nm - green) and LYIQNL (206 nm - blue) were plotted as a function of time, both in the presence (darker shades) and absence (brighter shades) of seeding crystals.

aromatic-aromatic interaction dominates the chiroptical spectrum after 6 h (a strong couplet at 225 nm and 237 nm, Fig. 5b green), showing that aromatic interactions play a key role in the aggregation mechanism of LYIQWL in water. Previous NMR measurements on the native monomeric E5 miniprotein (containing the LYIQWL sequence) showed that aromatic interactions (Trp - Tyr) play a key role in aggregation and that the spectral contribution is more dominant in the case of smaller oligomers. (Supplementary Fig. 2).

In contrast to the behaviour of the LYIQWL oligopeptide, LYIQNL self-arranges in the form of parallel β-sheets in water (Fig. 5d–f) suggested by the absence of the FTIR band characteristic of antiparallel β-sheets (1690 cm$^{-1}$). The IR spectrum also indicated the formation of additional H-bonds between the side-chain amide groups along the fibril axis, referred to as Gln and/or Asn ladders, which is consistent with the parallel orientation of the β-sheets (Table 2 and Supplementary Fig. 3).

Adding ethanol to the solvent system creates significant differences in the solution state structure of both systems as inferred from the ECD and FTIR spectra (Table 2 and Fig. 5). In contrast to pure water where LYIQWL forms antiparallel β-sheets (see FTIR data), the absence of the 1690 cm$^{-1}$ FTIR band suggests that if ethanol is present in the solution parallel, rather than antiparallel β-sheets form (Fig. 5j, k). In 10% ethanol amyloid formation of LYIQWL is perhaps slower, compared to that in water as the intensity of the ECD spectra are lower (Fig. 5h) with the strong transient π-π interaction missing. A different kind of aromatic interaction is indicated however by the ECD spectrum, which is consistent with the Tyr-Tyr interaction observed in the crystal structures. While LYIQNL assembles into parallel β-sheets stabilized by Gln and Asn ladders in 10 v/v% ethanol too, both spectroscopic methods indicate the conformational fixation of the Tyr side chains. This spectral feature was missing in the absence of ethanol.

AFM images showed fibrillar or nanocrystalline structures in all samples, in line with the spectroscopic results (Fig. 5c, f, i, l). Because LYIQWL readily crystallizes, microcrystal formation competes with fibril formation, resulting in small fibrils, while LYIQNL gives more developed, longer amyloid fibrils (nearly nanocrystals), especially in 10 v/v% ethanol. These AFM data suggest that a reduction in the polarity of the solvent could facilitate the maturation of amyloid fibrils.

In conclusion, both hexapeptides form amyloid assemblies in water and also in 10 v/v% ethanol, although characteristic differences – just as in case of their crystal structures - are already apparent in early stages of the association in the solution phase. In line with the crystallographic studies, the spectroscopic results also indicated that subtle changes in the solvent composition could result in changes in the evolving assemblies even as profound as a change in the relative orientation of the composing β-sheets. This might allow us to assign features of the B-type ECD spectrum of the parallel β-stranded amyloid nanostructure in solution with characteristic π → π* and n→π* transitions: an intense positive band at 208–210 nm and a slightly weaker negative band at 223-225 nm.

### Amyloid nucleation is catalysed by pre-grown, amyloid-like crystals

To elucidate the seeding potential of amyloid-like crystals and determine whether they can serve as compatible catalysts to enhance the conversion of unfolded monomers into β-stranded oligomers in solution, we performed ECD experiments in the presence and absence of seed crystal suspensions (Fig. 6). To minimize the effect of either the soluble oligomers present in the crystallization conditions or other components of the crystallization solutions (acetate buffer in case of LYIQNL), we extensively washed the crushed crystals and resuspended them in water.

At the start of the experiments, we observed spectra characteristic to unstructured monomers which could be measured also immediately after the addition of seed crystals (Supplementary Fig. 4). This showed that the added suspension did not contribute to the spectra.

However, within 15 min, the spectra of the crystal-seeded samples of LYIQWL (Fig. 6a) and LYIQNL (Fig. 6b) evolved into that of β-sheet secondary structure, whereas in the control samples, the β-type ordering was preceded by a lag phase of 90 and 45 min, respectively. Despite differences in the kinetics of amyloid formation in response to seeding, the CD signals of the differently treated peptide sample pairs decayed to the same endpoints overnight. This suggests that regardless of the presence or absence of seed crystals, amyloid aggregation results in the formation of larger systems, above the detection limit of ECD spectroscopy, eventually leading to precipitation. Nevertheless, the similar characteristics of the ECD spectra obtained with and without seeding suggest that structurally similar amyloid systems are formed, the addition of seed crystals only accelerates the process.

### Discussion

Only a few APR oligopeptides were reported so far that form different, polymorphic amyloid-like crystal structures (often using drastically different crystallization conditions to induce the changes), and apart from the current two, only four other sequences have been found to be able to self-assemble into crystals belonging to different topological classes (Supplementary Table 2). This however does not necessarily mean that

polymorphism is a rare occurrence among the amyloid-like crystals of APR segments; it might only be the result of there being few studies that were specifically focused on this question. Our results indicate that – at least in the case of the studied sequence motifs - the potential energy surface of APRs is composed of easily accessed, close lying states. We found that a simple change in crystallization temperature or modest changes of chemical composition of the solvent system (therefore the available interaction partners) can prompt formation of significantly different nanostructures: we identified four different polymorphs of the LYIQWL and two of the LYIQNL APR peptides. It is especially notable, that from identical solutions of LYIQWL – the APR of Tc5b, a truncated analogue of Exendin-4, a medication currently in use for the treatment of Type II Diabetes – both parallel and anti-parallel crystal forms could be obtained by simply changing the temperature at which crystallization was carried out. We also note that the lack of charged gatekeeper residues and their almost palindromic sequences might make the studied hexapeptides more amenable to form several different polymorphic structures.

Additives and co-solvents often used to crystallise macromolecules can influence both the structural features and the kinetic parameters of amyloid formation. The use of different solvent mixtures may therefore provide an opportunity to map the polymorphic space of an APR. Cofactors have been shown to stabilise the structure of several physiologically relevant amyloids formed from full-length proteins[55,56]. Therefore, solvent molecules in the crystal structures may also be involved in forming fibrils. Several oligopeptide amyloids were shown to bind small molecules and catalyse chemical reactions[57–59]. From this perspective, it is important to focus on the observation of small molecules incorporated into amyloid crystal structures, as these molecules may pinpoint potential binding surfaces of amyloid fibrils. The two LYIQWL structures grown from 10% ethanol in the presence or absence of residual TFA show that the solvent composition can have a profound effect on the topology of amyloid nanostructures and crystal structures. On the other hand, our ECD and FTIR results show that crystal structures containing incorporated solvent molecules might reflect associations already present in the solution phase.

It has been a long-standing question to what extent the topology of amyloid-like crystals resembles that of the amyloid fibrils or the still solubilized, initial oligomers that are formed from the same peptide. Our current spectroscopic results indicate similar solvent-dependent structural differences in solution phase to that of observed in the crystal structures. This suggests a degree of similarity in the topology (e.g., β-sheet orientation, hydrogen bonded interactions…) of amyloid-like crystals and the smaller oligomers in the solutions of the same peptide under similar experimental conditions. For example, our solution phase results on LYIQNL are consistent with the determined crystal structures, even though the crystal with a class 3 topology contained an ethanol molecule embedded in one of the steric zipper interfaces. However, it should be noted that in this crystal, the 'odd interface' (formed by Leu1, Ile3 and Asn5) is a dry one, that doesn't contain a solvent molecule, and therefore this molecular architecture may be stable even without the ethanol molecule present. The fact that seed crystals can be used to catalyse amyloid formation, which in turn can be detected by ECD spectroscopy, also seems to suggest some degree of structural similarity. We note however that our observations do not eliminate the possibility of the presence of polymorphs or a greater degree of structural heterogeneity in solutions.

With the wealth of structural data now appearing due to the advances of solid-state NMR and cryo-EM, it is becoming increasingly certain that polymorphism is an inherent property of physiologically relevant amyloid systems too and its pathophysiological significance is also being recognized[60–62], but the pool of obtainable amyloid polymorphs appears to be dependent on the primary sequence[38]. This was observed in our findings too. The W5N switch reduced the structural heterogeneity of the crystal-phase of LYIQNL as compared to that of LYIQWL, and influenced their topological preferences. The results also indicate that while some topologies are easily destabilized by environmental changes, certain others are able to withstand great solvent and temperature variations. We found, for example,

that the same class 8 crystals of LYIQWL emerge from solvent systems containing 30 v/v% acetonitrile, 1–20 v/v% ethanol – in the presence of TFA – or 10 v/v% acetone (all crystallization experiments and their results are compiled in Supplementary Table 3) suggesting that the stability of certain steric zipper interfaces far exceeds those of others – in case of a specific sequence. This idea was first proposed by Eisenberg and co-workers, the originators of the APR-approach, observing that although GNNQQNY – the APR of Sup35, the most widely studied prion model - is able to form two polymorphic crystals under similar crystallisation conditions, the steric zipper interfaces associated with the two crystal forms are essentially identical, with only the packing of the 'double layers' differing [24]. Furthermore, in most polymorphic crystal pairs, both polymorphs belong to the same class (Supplementary Table 2), i.e., the difference between the structures is only a shift in the β-sheets forming the zipper interface, with the relative orientation of the β-strands remaining identical.

Our results clearly showed that the substitution of Trp5 of LYIQWL with Asn resulted in a similarly amyloidogenic sequence. LYIQWL, as the APR of the E5 and Tc5b miniproteins[40] is embedded in an α-helical segment (Supplementary Fig. 5c)[6], despite its preference for β-sheet formation. To see if LYIQNL appears in any naturally occurring proteins we searched for this segment in the PDB database[63] using BLAST[64]. Two proteins have been found that contain this segment. In *Bacteroides thetaiotaomicron* Endo-4-O-sulfatase the LYIQNL hexapeptide is an integrated part of an antiparallel β-sheet (Supplementary Fig. 5a)[18], while in human Palmitoleoyl-protein carboxylesterase NOTUM the very same unit adopts an α-helical secondary structure (Supplementary Fig. 5b). Gatekeeper residues were also observed flanking the sequence in both proteins: an Asp follows it in the sulfatase (LYIQNLD[412]) and two Arg-s surround it in the carboxylesterase (R[284]LYIQNLGR[292]) (Supplementary Fig. 5), suggesting that LYIQNL might have been evolutionarily recognized as an APR and the incorporating proteins duly protected from its harm.

Originally 8 basic topologies were derived for crystal structures composed of linear β-strands, which together (with the later added additional two topologies) span the theoretically available conformational space of these systems. Among APR amyloid-like crystals, examples for 7 classes were previously obtained, but a crystal forming steric zippers of exclusively class 3 topology has not been previously encountered, even after the determination of well over 170 amyloid crystal structures. The crystals grown from LYIQNL thus provide the first example of a pure class 3 amyloid topology. That only one example of this topology has been seen so far, raises the question whether this arrangement is inherently less favourable compared to the other topologies.

Electrostatic repulsion between like-charged polypeptide chain-termini makes parallel arrangements in APR crystals less favourable[38] than the antiparallel β-sheet orientation. Both class 2 and class 3 assemblies are arranged as parallel oriented β-strands, both along and perpendicular to the main fibril axis (Fig. 7a), making electrostatic contributions in these classes of amyloid crystals the least favourable. MD simulations and the class 2 topology structures have previously shown that unfavourable electrostatic interactions can be compensated by favourable side-chain interactions in oligopeptide amyloids, both in the solution and in the crystalline phase[38]. The monoclinic crystal form of LYIQNL supports this conclusion by providing the previously missing example of the class 3 arrangement. By comparing the class 3 and class 4 structures formed by the same sequence, we found that in addition to favourable side-chain interactions, unfavourable electrostatic interactions are also mitigated by the larger distances between chain ends (Fig. 7b, c).

The uniqueness of the class 3 amyloid topology lies in the fact that each amino acid sidechain is surrounded by copies of itself both along the parallel β-strands that form β-layers, and in the perpendicular direction, within the dry-zipper interfaces. This requires a self-complementarity that is not easy to fulfil. Thus flexible sidechains that can self-interact are expected to potentiate a sequence toward this arrangement. Accordingly, we found strong and complex Asn5↔Asn5 and Gln4↔Gln4 interactions in the class 3 structure of LYIQNL (Fig. 3), as they form H-bond networks "with

**Fig. 7 | Interaction between charged termini in structures of LYIQNL. a** Schematic topological arrangement of class 2 and class 3 amyloids, both with alike charged N- and C-terminals. We have measured larger end-to-end distances between alike-charged chains both for class 3 (**b**) and class 4 (**c**) amyloid crystal structures of LYIQNL, as could have expected.

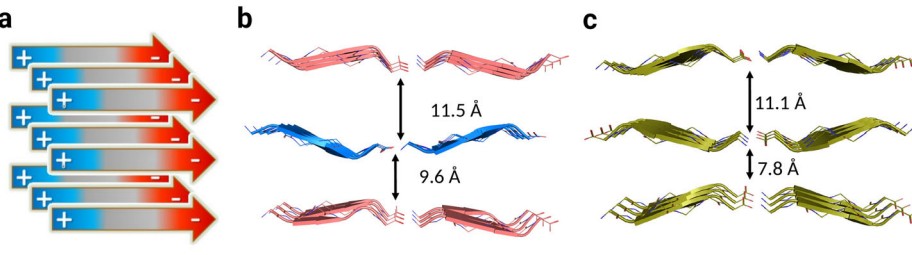

themselves" in all three directions. Note that at these connecting 3D junctions, the two sides of the dry zipper are closer than average. Finding a crystal which contains exclusively class 3 interfaces is made significant by demonstrating that all originally proposed theoretical amyloid-like crystal forms are accessible for APR peptides. Also, since the success of structure-based amyloid prediction algorithms depends on the diversity of amyloid structures determined by X-ray diffraction and cryo-EM methods[65], the first example of the hitherto elusive class 3 structure may be a step towards more accurate predictions and better success in the design of amyloid sequences with specified structural properties.

Altogether, the present work illustrates that – surprisingly - the structural complexity and keen sensitivity to environmental changes seen in case of physiological amyloids is already present in assemblies of the short APR segments, providing an unparalleled opportunity for the detailed study of the mechanisms, structural determinants, and sequence requirements of amyloid-polymorphism within clear and concise experimental settings.

## Materials and methods
### Peptide synthesis
Peptides were synthesised using in-house developed flow chemistry-based solid phase peptide synthesizer[66,67] following the standard Fmoc/tBu strategy. We used preloaded (containing the first C-terminal residue) Fmoc-AA-Wang TentaGel resins, and coupling reactions were performed in DMF at 80 °C under 70–90 bar pressure with OxymaPure®/DIC reagents. Oligopeptides were cleaved from the resin using a mixture of 2.5 v/v% triisopropylsilane, 2.5 v/v% water and 95 v/v% TFA at room temperature with stirring for 3 h. After evaporation of the TFA, the residue was precipitated in cold diethyl ether, washed three times with fresh ether and finally vacuum-dried. Peptides were purified by reverse-phase HPLC (Jasco LC-2000Plus HPLC system) using C18 column (Phenomenex® Kinetex 5 μm 150 × 21.2 mm,) and lyophilised. Analytical purity was confirmed by both analytical HPLC (Jasco LC-2000Plus HPLC system with Aeris™ 3.6 μm PEPTIDE XB-C18 LC Column 250 × 4.6 mm) and MS (Bruker AmaZon SL™ Electrospray ionization Ion Trap Mass Spectrometer). (Supplementary Table 4 and Supplementary Fig. 6).

### Sample preparation
LYIQWL and LYIQNL peptide samples were dissolved in 0.01 M HCl solution and lyophilized to remove residual TFA. (The intense vibration of the carboxyl group in TFA at 1673 cm⁻¹ overlaps with the Amide I band, making secondary structure analysis difficult[68].) After removing TFA from the samples, they were dissolved in bidestillated water at a concentration of 1.5 mg/mL. The pH of the peptide solution was determined by an Orion Star A211 pH meter (Thermo Scientific™) and the pH was adjusted to 3.8 using 0.1 M HCl and 0.1 M NaOH solutions. During time-dependent measurements (from 0 h to 48 h), peptide solutions were stirred with a magnetic stirrer at 500–600 rpm, incubated at 37 °C.

### Fourier-transform infrared spectroscopy (FTIR) measurements
FTIR measurements were carried out on a Bruker (Billerica, MA, USA) Equinox 55 spectrometer equipped with a bio-ATR (attenuated total reflectance) cell, where the internal reflection element is made of a ZnSe crystal. The photoelastic modulator (ZnSe) was set to 1600 cm⁻¹ and an

optical filter with a transmission range of 1900–1200 cm⁻¹ was used to increase sensitivity in the amide I-II spectral range. The mercury-cadmium-telluride (MCT) detector was cooled with liquid nitrogen. Spectra were obtained by averaging 128 scans in the range of 4000 cm⁻¹ to 850 cm⁻¹ with a resolution of 4 cm⁻¹, using an aperture of 3000 microns. All absorption spectra were solvent-subtracted. After atmospheric compensation, deconvolution of the spectra was carried out using the Curve Fit procedure of the OPUS 6.0 software. FTIR, ECD and AFM measurements were performed simultaneously from the same stock solutions.

### ECD measurement
The UV-CD measurements were performed with JASCO (Tokyo, Japan) J-810 and J-1500 spectropolarimeters, equipped with a Peltier. Each spectrum was the average of five scans collected in the far UV (185–260 nm) range with a 0.1 mm path length quartz cuvette cell. The following settings were used throughout the measurements: a temperature control system at room temperature, a bandwidth of 1 nm, a step size of 0.2 nm, a response time of 4 s and a scan rate of 50 nm/min. All spectra were corrected by subtracting the solvent spectrum acquired under identical conditions. All CD data were processed from mDeg to mean residue ellipticity (deg cm² dmol⁻¹) using the Spectra Analysis function of Jasco Spectra Manager, to account for the concentration differences.

The molar concentration of the prepared samples was determined using a NanoDrop Lite spectrophotometer (Thermo Scientific™), using the molar extinction coefficients ($\varepsilon_{Tyr}$ = 1280 M⁻¹cm⁻¹ and $\varepsilon_{Trp}$ = 5690 M⁻¹cm⁻¹) of the model compounds Gly-L-Tyr-Gly and N-Acetil-L-Trp-amide, respectively at 280 nm[69]. FTIR, ECD and AFM measurements were performed simultaneously from the same stock solutions.

### Seeding experiments
Crystals of LYIQWL and LYIQNL were grown under identical conditions as crystals used for X-ray diffraction measurements. The seed crystal suspension was prepared by following the procedure described by Sawaya et al.[24]. Briefly, crystals were sonicated for 3–5 min in the crystallization buffer, then centrifuged for 10 min at 20,000 × g. After removing the supernatant, crystals were resuspended in the buffer used for spectroscopic measurements and incubated at 37 °C for 1 h with shaking at 1000 rpm. After the incubation, the crystals were centrifuged again for 10 min at 20,000 × g, the supernatant were removed and the pellet was resuspended in the buffer used for spectroscopic measurements.

To be able to observe the lag phase of amyloid formation, a low (0.18 mg/ml) peptide concentration was used for these experiments. The samples were prepared in bidistilled water, filtered immediately after dissolution through a 0.45 μm pore size filter and their pH was adjusted to 4.2. Spectra were recorded parallel with and without the addition of crystals. 10 μl of the seed crystal suspension was added to the peptide solutions at the start of the time-dependent measurement, after recording an ECD spectrum without the presence of crystals. Samples were incubated at 37 °C with gentle agitation at 400 rpm.

## Table 3 | Data collection and refinement statistics

| | LYIQNL monoclinic crystal form (class 3) | LYIQNL orthorhombic crystal form (class 4) | LYIQWL |
|---|---|---|---|
| **Data collection** | | | |
| Space group | P2₁ | P2₁2₁2₁ | P2₁ |
| Cell dimensions | | | |
| *a, b, c* (Å) | 4.840, 42.425, 22.256 | 4.848, 20.005, 42.650 | 4.874, 42.270, 21.442 |
| α, β, γ (°) | 90.00, 94.57, 90.00 | 90.00, 90.0, 90.00 | 90.00, 90.0, 90.00 |
| Resolution (Å) | 19.66–1.70 (1.76–1.70)[a] | 18.11–1.55 (1.61–1.55) | 19.05–1.50 (1.55–1.50) |
| $R_{meas}$ | 0.147 (0.630) | 0.160 (0.414) | 0.101 (0.567) |
| $I/\sigma I$ | 5.9 (1.3) | 7.4 (3.5) | 7.5 (1.3) |
| Completeness (%) | 99.30 (92.13) | 94.01 (68.83) | 97.93 (88.00) |
| Redundancy | 2.83 (1.82) | 4.83 (4.11) | 3.28 (1.40) |
| **Refinement** | | | |
| Resolution (Å) | 19.66–1.70 | 18.11–1.55 | 19.05–1.50 |
| No. reflections | 982 | 704 | 1362 |
| $R_{work}/R_{free}$ | 0.1546/0.1893 | 0.2344/0.2437 | 0.1357/0.1737 |
| No. atoms | | | |
| Protein | 121 | 51 | 138 |
| Solvent | 14 | 0 | 1 |
| *B*-factors | | | |
| Protein | 17.53 | 10.85 | 10.71 |
| Solvent | 35.16 | - | 19.86 |
| R.m.s. deviations | | | |
| Bond lengths (Å) | 0.004 | 0.013 | 0.006 |
| Bond angles (°) | 0.715 | 1.034 | 1.158 |
| PDB code | 8QWV | 8QWU | 8QWW |

[a]Values in parentheses are for highest-resolution shell.

### Energy calculations

Energies of formation of double layers along given interfaces were calculated using atomic solvation parameters as described by Hughes et al.[49]. Briefly, double layers consisting of 2 × 3 (for parallel structures) or 2 × 6 (for antiparallel structures) were built using crystal symmetry in PyMOL. The solvent exposed surface area were calculated for each atom in the double layer and also in individual β-sheets. The two values were subtracted to get the change in the exposed surface area upon double layer formation. The changes in exposed surface area of each atom were multiplied with atomic solvation paramaters[49] to estimate the free energy contribution of that atom. These values were then summed for each atom of the central strand(s) and divided by the number of strands used for the calculation to obtain the estimated free energy of formation of a double layer.

For calculating the energies of formation from peptide monomers we used the same systems containing 2 × 3 or 2 × 6 strands. We calculated the energies using the method proposed by Sawaya et al.[50] with the software made available by the Eisenberg group (version 7.2e, https://doi.org/10.5281/zenodo.6321286).

### AFM measurement

AFM measurements were performed on a FlexAFM microscope system (Nanosurf AG, Liestal, Switzerland), operating in dynamic mode controlled by the Nanosurf control software C3000 version 3.10.4. The measurements were performed with Tap150GD-G cantilevers from BudgetSensors Ltd. (Sofia, Bulgaria) with a nominal tip radius of less than 10 nm. Prior to data collection, low resolution preliminary screening was performed to avoid significant height variations in the surface topology and to detect fibril-like structures. Data were collected from multiple locations within a single sample. Images were taken near densely packed aggregates on the surface using a window size of 10 × 10 µm and a resolution of 512 pixels per line. In some cases, data were collected from specific regions of interest with higher resolution. The AFM data were processed, and images were generated using Gwyddion 2.62 software. FTIR, ECD and AFM measurements were performed simultaneously from the same stock solutions.

### Crystallographic study

*LYIQNL, monoclinic crystal form*: the peptide was dissolved in 0.1 M acetate buffer, pH 4.80 at 0.15 mg/mL concentration in a small vial. The vial not fully closed with its cap was placed in a larger vial filled with 60 v/v% ethanol, 0.1 M acetate buffer, pH 4.80 and left at 20 °C. Thin needles appeared in a few days as ethanol evaporated into the smaller vial.

*LYIQNL, orthorhombic crystal form*: the peptide was dissolved in 0.1 M acetate buffer, pH 4.80 at 0.12 mg/mL concentration and incubated at 37 °C. Thin needle-like crystals appeared in a few weeks.

*LYIQWL*: The same re-lyophilized sample was used for crystallization that we used for spectroscopic studies. The peptide was dissolved in 10 v/v% ethanol at 0.5 mg/ml concentration and incubated at 37 °C. Needle-like crystals appeared in a cluster after a few weeks.

Single crystal X-ray diffraction data were collected at 100 K on a Rigaku XtaLab Synergy-R diffractometer using Cu Kα radiation. The CrysAlisPro v171.42.58a software (Oxford Diffraction/Agilent Technologies UK Ltd, Yarnton, England) was used for data collection and data reduction.

The phase problem was solved by molecular replacement using a polyalanine β-strand model generated from the previously solved structure of LYIQWL (PDB code: 8ANM) or 4–5 residue long ideal β-strands. Phaser[70] from the Phenix package[71] was used for structure solution. Manual model building was carried out in COOT[72], the resulting models were refined using Phenix.refine[73] and Buster[74]. The side chain of Leu6 was left out from the model of the orthorhombic crystal form of LYIQNL due to disorder. The φ/ψ angles of all models fall within the favoured regions of the Ramachandran plots. Data collection and refinement statistics are compiled in Table 3. Electron density maps around the asymmetric unit of the crystals can be seen on Supplementary Fig. 7.

### Statistics and reproducibility

AFM images were taken from random locations on the surfaces. Low-resolution pre-screenings were conducted before data collection to prevent significant height variations in surface topology and to identify fibril-like formations. Data collection was performed once from various locations within a single sample.

Growth of needle shaped crystals were observed in several conditions similar to the reported optimal crystallization conditions (e.g., slightly different peptide or ethanol concentrations). The reported conditions reproducibly resulted in crystal growth using different batches of the peptides. X-ray diffraction measurements were performed only from crystals grown in the optimal conditions. Typically several crystals were screened from the same crystallization experiments, yielding the same unit cell parameters, then data was collected from the best crystals.

ECD and IR measurements were typically performed once, except for ECD measurements in water, which were repeated for seeding experiments. These repeated measurements (from different batches of the peptides) resulted in identical spectra shapes.

### Reporting summary

Further information on research design is available in the Nature Portfolio Reporting Summary linked to this article.

### Data availability

Novel crystal structures have been deposited in the PDB database under accession codes 8QWV, 8QWU and 8QWW. Previously published crystal structures of LYIQWL can be obtained from the PDB database under

**Article**

accession codes 8ANM, 8ANI and 8ANG. Structures of *Bacteroides the-taiotaomicron* Endo-4-O-sulfatase, human Palmitoleoyl-protein carboxylesterase NOTUM and Tc5b were obtained from the PDB database with accession codes 6S21, 6ZYF and 1L2Y, respectively. Processed ECD and IR data used for the generation of graphs are provided in Supplementary Data 1. Raw spectroscopic data and unprocessed AFM images will be shared upon request by contacting the corresponding author.

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

## Acknowledgements

We are grateful to Veronika Harmat, Nóra Taricska and Viktor Farkas for critical insights during discussions. The crystallographic study was supported by project No. VEKOP-2.3.3-15-2017-00018 of the European Union and the State of Hungary, co-financed by the European Regional Development Fund. Project no. 2018-1.2.1-NKP-2018-00005 has been implemented with the support provided from the National Research, Development, and Innovation Fund of Hungary, financed under the 2018-1.2-1-NKP funding scheme (HunProtExc). This work was completed in the ELTE Thematic Excellence Programme supported by the Hungarian Ministry for Innovation and Technology. Project number RRF-2.3.1-21-2022-00015 is implemented with the support of the European Union's Recovery and Resilience Instrument (PharmaLab). Supported by the Ministry for Innovation and Technology from the Hungarian NRDI Fund (2020-1.1.6-JÖVŐ-2021-00010). The scientific work publicised in this article was reached with the sponsorship of Gedeon Richter Talentum Foundation in framework of Gedeon Richter Excellence PhD Scholarship of Gedeon Richter. The grant was awarded to Fruzsina Bencs.

## Author contributions

Zs.D., D.K.M. and A.P. designed and coordinated the project. Zs.D. crystallized the peptides, collected and processed diffraction data, carried out structure solution and analysed the structural data. F.B. synthesised and purified the peptides, measured FTIR and ECD spectra, processed and analysed the spectroscopic data. D.H. measured ECD spectra, collected and processed the AFM data. The manuscript was written by Zs.D., F.B., D.K.M. and A.P. and all authors contributed to it. The funding and instrumental background was provided by A.P.

## Funding

## Competing interests

The authors declare no competing interests.
