## [Peer Review File · Communications Biology]

Reviewers' comments:

Reviewer #1 (Remarks to the Author):

Perczel and coworkers describe different polymorphs of two hexapeptides (LYIQWL and the variant LYIQNL), including the elusive class 3 type which is rarely found due to the high degree of self-complementarity required to form this type of structure. In the present study, that structure is one of several adapted by LQIQNL, depending on solvent conditions. The work is well carried out though in its present form somewhat phenomenological and could benefit from additional steps to make further generalizations (see below).

General points:

1. I miss the rationale for making the LYIQNL variant. Did the authors try many different substitutions of Trp5?
2. "The side chain of Tyr2 forms H-bonds with chain termini". Show these H-bonds in Fig. 4c (currently unclear).
3. "In this arrangement, hydrophobic residues (Leu and Ile) form a cluster". Not clear – just sideways with each other? Is that a cluster? I assume there is no cluster with Gln on the other strand?
4. The Discussion about the relative stabilities of different beta-strand classes has many fine points but could benefit from attempts to do either MD simulations of different topologies or (more tractably) static calculations of the various structures (e.g. with FoldX or the Eisenberg/Sawaya approach described in <https://pubmed.ncbi.nlm.nih.gov/34534463/>). I appreciate that the role of cosolvents in the crystallization processes may complicate calculations but this is a good starting point for an even more nuanced discussion.

Reviewer #2 (Remarks to the Author):

The manuscript uses single-crystal X-ray diffraction spectroscopy to investigate molecular structures of the peptides LYIQNL and LYIQWL. While both peptides form beta-sheet-rich assemblies at a range of conditions, the authors show interesting molecular differences, solely depending on the exact conditions or a single amino acid substitution. The set of structures includes the previously elusive type 3 interface. The authors then compare the insight from the crystal structures to samples aggregated in more common, amyloid-forming conditions using IR, CD, and AFM.

The "missing" type 3 interface is an excellent addition to peptide amyloid structures from biologically derived sequences.

The authors explore several conditions and show how small changes in both conditions and sequence can lead to different structures and polymorphisms. The authors further highlight the surprising structural complexity of relatively short peptide amyloids and their use for understanding the polymorphism of larger systems.

The structural work on single-crystal X-ray is sound.

However, there are also the following concerns/issues/problems:

(1) The crystallization conditions are clearly different from the conditions used for spectroscopic (pH, buffer system (if any), time of amyloid/crystal growth). Given that, and given the limited structural information from CD, AFM, and IR (also regarding potential polymorph mixtures), it seems stretched to claim structural similarities between them.

(2) How the spectroscopic samples (possibly amyloid fibrils including a twist) differ from the crystals is unclear. At least some negative stain TEM images of the former would be a nice addition to the paper. Or/And the observation of growth of crystal end of fibrils as demonstrated by Eisenberg or/and seeding potential of the crystals.

(3) The part titled "LYIQNL and LYIQWL segments are shielded in proteins from amyloid formation." These are interesting observations. However, no new data is provided; therefore, a shortened version of the paragraph would better fit into the introduction(or possibly the discussion) rather than the results.

Comments on figures:

Fig 1: It could be more visually appealing, but it conveys the message.

Fig 2: The side chains are hardly visible and too small/thin. There are too many layers depicted (it is obvious that there are many layers, and the interfaces are what is important). It seems that the color code given in (e) should correspond to (a-d), but it is hard to see which one is which, both in (a-d) and in the overlay in (e).

Fig 3: Similar issues as in Fig 2

Fig 4: This figure again gets the message across but could be more visually appealing/elegant. Perhaps do not include the non-colored parts of the structure or display only one layer.

Fig 5: Summarizes the results well.

Fig 6: It is questionable if this figure is necessary, as it does not display any new data.

Fig 7: Typo in figure caption?

Answers to the Reviewers' comments

Reviewer 1

We thank Reviewer 1 for the thorough review of our manuscript as well as the helpful suggestions!

Perczel and coworkers describe different polymorphs of two hexapeptides (LYIQWL and the variant LYIQNL), including the elusive class 3 type which is rarely found due to the high degree of self-complementarity required to form this type of structure. In the present study, that structure is one of several adapted by LQIQNL, depending on solvent conditions. The work is well carried out though in its present form somewhat phenomenological and could benefit from additional steps to make further generalizations (see below).

General points:

1. I miss the rationale for making the LYIQNL variant. Did the authors try many different substitutions of Trp5?

In our previous study (<https://pubmed.ncbi.nlm.nih.gov/37528104/>) we observed that LYIQWL is able to form at least 4 polymorphic amyloid-like crystal structures composed of both parallel and antiparallel β -sheets. We also found that while the overall conformation of the peptide monomers was very similar, side chains – especially that of Trp5 – could adopt different conformations. We asked the question whether the bulky nature of Trp contributed to the high variability of polymorphs we observed. As a continuation of our previous work, here we focused on the effects of replacing a bulky, hydrophobic residue with a smaller, less flexible, hydrophilic residue which was also shown to favor amyloid formation. Furthermore, due to their preference for participating in hydrogen bonded ladders, Asn (and Gln) residues favor parallel β -sheets. As we observed both parallel and antiparallel structures of LYIQWL, we were also interested how the incorporation of a further “parallel β -sheet favoring residue” would change the pool of possible polymorphs.

We agree with the reviewer that a brief explanation behind the Trp5Asn mutation would improve the clarity of our manuscript, therefore the following text was inserted at the end of the introduction (after line 99):

“In these structures the side chain of Trp5 adopted a variety of different conformations, suggesting that this residue might play a role in the amyloid polymorphism of LYIQWL. We asked the question, whether a single point mutation of Trp5 would be enough to make the parallel orientation of the β -sheets more favorable. We selected asparagine, as its side chain is less bulky than tryptophan and it was shown to stabilize parallel β -sheets by forming hydrogen-bonded ladder motifs⁴¹. In the present work we therefore synthesized the Trp5Asn mutant of LYIQWL...”

2. “The side chain of Tyr2 forms H-bonds with chain termini”. Show these H-bonds in Fig. 4c (currently unclear).

We modified both Fig. 3 and Fig. 4. On the new Fig 3d these atoms are more visible and on Fig 4b we now explicitly show the mentioned hydrogen bonds.

Fig. 3.

Fig. 4.

3. “In this arrangement, hydrophobic residues (Leu and Ile) form a cluster”. Not clear – just sideways with each other? Is that a cluster? I assume there is no cluster with Gln on the other strand?

We agree that the wording could be clearer in this case. By cluster we meant a spatial gathering of hydrophobic side chains and we intended to emphasize that both in the class 3 and the class 4 structures of LYIQNL a remarkably similar hydrophobic center could form. We have rephrased the sentence in question as follows (line 198):

“Similarly to the class 3 structure, hydrophobic residues (Leu and Ile) form a hydrophobic niche in this arrangement too, while polar side chains of Asn and Gln stabilize the interface enforced by H-bonds (Fig. 3f and Fig. 4d)”

4. The Discussion about the relative stabilities of different beta-strand classes has many fine points but could benefit from attempts to do either MD simulations of different topologies or (more tractably) static calculations of the various structures (e.g. with FoldX or the Eisenberg/Sawaya approach described in <https://pubmed.ncbi.nlm.nih.gov/34534463/>). I appreciate that the role of cosolvents in the crystallization processes may complicate calculations but this is a good starting point for an even more nuanced discussion.

Thank you for the suggestion!

We calculated the energy of formation of a hypothetical “double layer” along given interfaces using atomic solvation parameters as described by Hughes et. al.

(<https://pubmed.ncbi.nlm.nih.gov/29439243/>) and also the energy of formation of the same system from monomeric peptides using the Eisenberg / Sawaya approach (<https://pubmed.ncbi.nlm.nih.gov/34534463/>) We found that different polymorphs of the same sequence are indeed close in energy, which explains why it is so easy to change the polymorph observed with only subtle environmental changes.

We think that these results are indeed a valuable addition to the manuscript, therefore we included the following paragraphs:

“In order to gain a better understanding of the polymorphic nature of the two sequences, we studied the energetics of their amyloid formation using the determined crystal structures. We calculated both the energy gain upon formation of a “double layer” along a given interface from individual β -sheets using atomic solvation parameters⁴⁹ and the free energy of formation of the same “double layer” from monomeric peptide chains using the method proposed by Eisenberg et. al.⁵⁰ (Table 1).

Table 1. Energetics of formation of a hypothetical double layered amyloid structure along given interfaces found in the crystal structures

Sequence	Polymorph	Interface	E_{form} (from sheets) [kcal · mol ⁻¹ / strand]	E_{form} (from monomers) [kcal · mol ⁻¹ / strand]
LYIQWL	8ANM (class 8)	S1	-0.85	-5.19
	8ANI (class 8)	S1	-1.54	-4.58
	8ANG (class 1)	S1	-1.66	-5.31
	8QWW (class 4)	S1	-1.30	-4.56
		S2	-1.37	-4.63
LYIQNL	8QWV (class 3)	S1	-1.17	-3.56
		S2	-0.76	-3.21
	8QWU (class 4)	S1	-1.23	-3.80

The energy gain upon formation of a hypothetical double layer from individual β -sheets is comparable to that of found for stable steric zippers (median = -1.417 kcal/mol/strand⁴⁹) and significantly higher than that of found for LARKS (median = -0.443 kcal/mol/strand⁴⁹) by Hughes et. al. Lower values were found in case of 8ANM where a small interface with small buried area formed and in S2 from 8QMV, where an ethanol molecule is part of in the interface.

Furthermore, both measures – but especially the energy of formation from monomers – show slightly more favourable energetics in case of LYIQWL than of LYIQNL, while the energy difference between different polymorphs of the same peptide sequence is decidedly lower. Since the atomic solvation parameters were optimized using water as the solvent, while several of the polymorphs discussed here were observed only in the presence of ethanol, some error might have been introduced into our estimates, but the results clearly reflect that polymorphism is simply enabled by the possibility of forming assemblies of similar energies with different structural characteristics.”

Reviewer 2

We thank Reviewer 2 for the critical review of our manuscript and the helpful comments suggestions!

The manuscript uses single-crystal X-ray diffraction spectroscopy to investigate molecular structures of the peptides LYIQNL and LYIQWL. While both peptides form beta-sheet-rich assemblies at a range of conditions, the authors show interesting molecular differences, solely depending on the exact conditions or a single amino acid substitution. The set of structures includes the previously elusive type 3 interface. The authors then compare the insight from the crystal structures to samples aggregated in more common, amyloid-forming conditions using IR, CD, and AFM.

The "missing" type 3 interface is an excellent addition to peptide amyloid structures from biologically derived sequences.

The authors explore several conditions and show how small changes in both conditions and sequence can lead to different structures and polymorphisms. The authors further highlight the surprising structural complexity of relatively short peptide amyloids and their use for understanding the polymorphism of larger systems. The structural work on single-crystal X-ray is sound.

However, there are also the following concerns/issues/problems:

(1) The crystallization conditions are clearly different from the conditions used for spectroscopic (pH, buffer system (if any), time of amyloid/crystal growth). Given that, and given the limited structural information from CD, AFM, and IR (also regarding potential polymorph mixtures), it seems stretched to claim structural similarities between them.

In the case of LYIQWL the crystallization conditions are essentially identical to the conditions used for spectroscopic measurements (water and 10 v/v% ethanol). LYIQWL crystals even grew from ECD samples (from water) with identical structure to those grown for X-Ray diffraction measurements.

In the case of LYIQNL, diffraction quality crystals were indeed grown in the presence of 0.1M acetate buffer, whereas no buffer was used during spectroscopic measurements. We note that the first crystals grew from solutions without buffers, however, we added buffers during optimization. As explained in more detail in the answer of the next question, we assessed the seeding potential of these crystals in ECD experiments and found that – similarly to crystals of LYIQWL - they indeed catalyzed the amyloid formation.

We agree that the oligomers or short fibrils present in solutions used for the spectroscopic experiments are more than likely to differ structurally from the crystals. We also agree that polymorphism or structural heterogeneity could not be ruled out either. Although the structural resolution of these methods is indeed limited, we believe that certain aspects of the studied systems can be compared to the crystal structures. For instance, IR spectroscopy indicated that ethanol induces an antiparallel – parallel switch in case of amyloid structures formed by LYIQWL. The same effect could be observed in the crystal structures, indicating a similar tendency in solution and crystalline states. By structural similarity we meant similarities of these “low resolution” characteristics of the systems and did not intend to claim more than that.

We reviewed the manuscript and rephrased the sentences which could suggest a higher degree of structural similarity than what can be proved by the techniques we used.

(2) How the spectroscopic samples (possibly amyloid fibrils including a twist) differ from the crystals is unclear. At least some negative stain TEM images of the former would be a nice addition to the paper. Or/And the observation of growth of crystal end of fibrils as demonstrated by Eisenberg or/and seeding potential of the crystals.

Thank you for the suggestion!

Perhaps the most challenging aspect of structural studies of amyloid systems is the fact that with different methods we obtain information about systems of different sizes (eg. oligomers in solutions vs. nearly infinite number of ordered molecules in the crystal phase) which could in turn show different structural features.

In order to study the seeding potential of the crystals, we prepared seed crystal suspensions according to the protocol described by Sawaya et. al. (<https://pubmed.ncbi.nlm.nih.gov/17468747/>) and used them in ECD experiments.

For these experiments we used a lower peptide concentration in order to be able to measure a sufficiently long lag phase, which would make the assessment of any effect of the crystals easier. We selected two systems for these studies: LYIQWL in water seeded with crystals grown under identical conditions, and LYIQNL in water seeded with crystals grown in 0.1M acetate buffer. Seed crystals were washed into water in both cases to eliminate any possible effect of the crystallization buffer.

We found that the addition of seed crystals shortened the lag phase of amyloid formation in both systems considerably: from 90 min to <15 min and from 45 min to <15 min in the case of LYIQWL and LYIQNL, respectively.

These results suggest that the crystals possess sufficient structural similarity to the solution phase oligomers observable with ECD spectroscopy to act as seeds and catalyze their formation. This was true even for the crystals of LYIQNL, which were grown in acetate buffer instead of water, suggesting that the presence of acetate buffer does not alter the main structural characteristics of these amyloid systems.

We also tried to study the seeding potential of the crystals with ThT assay, but with this technique we could not observe a lag phase of the process, therefore we could not assess the effect of seeding.

We think that these results are indeed a valuable addition to the data presented in the manuscript, therefore we inserted the following paragraphs and figure:

“To elucidate the seeding potential of amyloid-like crystals and determine whether they can serve as compatible catalysts to enhance the conversion of unfolded monomers into β -stranded oligomers in solution, we performed ECD experiments in the presence and absence of seed crystal suspensions (**Fig. 6**). To minimize the effect of either the soluble oligomers present in the crystallization conditions or other components of the crystallization solutions (acetate buffer in case of LYIQNL), we extensively washed the crushed crystals and resuspended them in water.

At the start of the experiments, we observed spectra characteristic to unstructured monomers which could be measured also immediately after the addition of seed crystals (**Supplementary Fig. 7**). This showed that the added suspension did not contribute to the spectra.

However, within 15 minutes, the spectra of the crystal-seeded samples of LYIQWL (**Fig. 6a**) and LYIQNL (**Fig. 6b**) evolved into that of β -sheet secondary structure, whereas in the control samples, the β -type ordering was preceded by a lag phase of 90 and 45 minutes, respectively. Despite differences in the kinetics of amyloid formation in response to seeding, the CD signals of the differently treated peptide sample pairs decayed to the same endpoints overnight. This suggests that regardless of the presence or absence of seed crystals, amyloid aggregation results in the formation of larger systems, above the detection limit of ECD spectroscopy, eventually leading to precipitation. Nevertheless, the similar characteristics of the ECD spectra obtained with and without seeding suggest that structurally similar amyloid systems are formed, the addition of seed crystals only accelerates the process.

Figure 6. The effect of seeding with crystals to the kinetics of amyloid formation monitored by ECD spectroscopy. The secondary structure conversions in peptide solutions of LYIQWL (**a**) and LYIQNL (**b**) were monitored over a day, focusing on the nucleation phase with frequent sampling. Intensity values corresponding to wavelengths characteristic of the β -sheeted state for LYIQWL (235 nm - green) and LYIQNL (206 nm - blue) were plotted as a function of time, both in the presence (darker shades) and absence (brighter shades) of seeding crystals.”

We also present the complete ECD data set in the supplementary information:

“Supplementary Figure 7. Amyloid formation monitored by CD spectroscopy in presence and absence of seed crystals. In the right column, the recorded CD spectra reveal the secondary structure conversions as a function of time for LYIQWL/LYIQNL in the absence (a and e) and presence (c and g)

of added seeding crystals, respectively. We highlight the corresponding changes in the CD intensity measured at characteristic wavelengths of LYIQWL (**b,d**) and LYIQNL (**f,h**).”

(3) The part titled "LYIQNL and LYIQWL segments are shielded in proteins from amyloid formation." These are interesting observations. However, no new data is provided; therefore, a shortened version of the paragraph would better fit into the introduction (or possibly the discussion) rather than the results.

We moved the general points of this section to the introduction and the points specific to the systems studied in this work to the discussion.

Comments on figures:

Fig 1: It could be more visually appealing, but it conveys the message.

Fig 2: The side chains are hardly visible and too small/thin. There are too many layers depicted (it is obvious that there are many layers, and the interfaces are what is important). It seems that the color code given in (e) should correspond to (a-d), but it is hard to see which one is which, both in (a-d) and in the overlay in (e).

We modified the figure to include only the layers necessary to show all interfaces present in the crystals. In the new figure we show side chains as sticks in order to make them more visible. The structures in (e) are also shown as sticks in order to make them and the colors more visible.

Fig 3: Similar issues as in Fig 2

We modified the figure similarly to Fig. 2.

Fig 4: This figure again gets the message across but could be more visually appealing/elegant. Perhaps do not include the non-colored parts of the structure or display only one layer.

We modified the figure to include only the relevant parts of the structure.

Fig 5: Summarizes the results well.

Fig 6: It is questionable if this figure is necessary, as it does not display any new data.

As we distributed the contents of this section between the introduction and the discussion, we also removed this figure from the manuscript. However, we think that it can be useful to demonstrate our points in the discussion, therefore we included it in the Supplementary Information file.

Fig 7: Typo in figure caption?

Thank you for noticing it! It has been corrected.

REVIEWERS' COMMENTS:

Reviewer #1 (Remarks to the Author):

I appreciate the authors' efforts to address my concerns and find them satisfactory and a nice improvement on the manuscript. No more questions from me.

Reviewer #2 (Remarks to the Author):

The manuscript is much improved (including the quality of the figures) and the answers of the reviewer have been addressed adequately.